# Correlation between Soil Bacterial Community Structure and Soil Properties in Cultivation Sites of 13-Year-Old Wild-Simulated Ginseng (*Panax ginseng* C.A. Meyer)

**Kiyoon Kim, Hyun Jun Kim, Dae Hui Jeong, Jeong Hoon Huh, Kwon Seok Jeon and Yurry Um ***

Forest Medicinal Resources Research Center, National Institute of Forest Science, Yongju 36040, Korea; kky0607@korea.kr (K.K.); mind4938@korea.kr (H.J.K.); najdhda@korea.kr (D.H.J.); mdgs3275@korea.kr (J.H.H.); jks2029@korea.kr (K.S.J.)
* Correspondence: urspower@korea.kr

**Abstract:** Soil properties are one of the major factors determining the growth of vegetation. These properties drive the selection of the dominant bacterial community profiles, which eventually determines the soil quality and fertility. The abundance of preferential bacterial community assists in better productivity of a particular type of vegetation. The increasing focus on the health and well-being of the human population has resulted in a shift in paradigm to concentrate on the cultivation of medicinal plants such as Wild-simulated ginseng (WSG). These plant species take a long time for their growth and are generally cultivated in the mountainous forest trenches of Far East countries like South Korea. This study was conducted to decipher the bacterial community profiles and their correlation with soil chemical properties, which would give a broader idea about the optimum growing conditions of such an important medicinal plant. The important edaphic factor determined in this study was the soil pH, which was recorded to be acidic in all the studied cultivation sites. In agreement with the edaphic factor, the relative abundance of *Acidobacteria* was found to be highest as this phylum prefers to grow in acidic soils. Moreover, the total organic matter, total nitrogen and cation exchange capacity were found to be significantly correlated with the bacterial community. Hence, these results will help to identify the suitable cultivation sites for WSG and increase the productivity of these medicinal plants.

**Keywords:** wild-simulated ginseng; *Panax ginseng* C.A. Meyer; soil bacterial community; soil property; correlation analysis

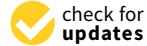

## 1. Introduction

Wild-simulated ginseng (WSG) belongs to the Araliaceae family, and it is also known as *Panax ginseng* C.A. Meyer [1]. It is mostly grown through artificially sowing the seeds or transplanting of seedlings in a mountainous area by the Korean Forest Service (KFS) [2]. In Korea, the WSG is defined as a kind of ginseng produced without the use of any artificial facilities, and the West Virginia legislature in the United States defined it as the ginseng grown in theforest without the use of any weed, disease or pest control agents. [3].

Soil microbes present in the rhizosphere have symbiotic relations with plants, and they can contribute to plant growth through decomposition of organic matter, nutrient (carbon, nitrogen and inorganic elements) cycling, removal of pollutants and supplying of nutrients to plants, and they play an important role in determining soil quality and productivity [4,5]. The recent develop in culture-independent methods has made it convenient to study microbial diversity and predict key functional traits of soil microbiota [6]. Myriad environmental factors can affect the soil properties and in turn tweak diversity and composition of soil microbiota [7,8]. Therefore, studying the correlation of environmental factors and soil microbial community is very important [9–14].

The recent focus on health and immunity has enhanced the interest in organic agriculture and exploitation of soil microbes for improvement in quality, and productivity of

medicinal crops is pacing up [1,15]. Ginseng (*Panax ginseng*) is a representative medicinal crop used in Far East countries, and there has been increasing interest in studying the soil microbial community present in their cultivation fields [16,17]. In addition, there has been a growing interest in studying the soil microbial communities based on the changes in forest environments [18–20]. However, the correlation between soil microbial communities and edaphic factors exerted on medicinal crops growing in forest is insufficiently studied.

The correlation of the soil microbial community with the edaphic factors for cultivation of WSG is important, as it is cultivated in the mountainous trench for a long period of time (~7–15 years) without the use of any pesticide or chemical fertilizers [21]. Thus, before cultivating WSG, it is necessary to investigate the suitability of cultivation by analyzing the edaphic factors of the site, such as soil properties and soil microbial communities [3]. Hence, the aim of this study was to investigate the correlation between soil properties and soil bacterial communities in different cultivation sites of WSG grown for 13 years in the forest ecosystem.

## 2. Materials and Methods

### 2.1. Study Area and Soil Sample Collection

A total of 9 different cultivation sites of WSG were chosen randomly in South Korea, and the details of the study area and the sampling sites are shown in Figure 1. Both rhizospheric and non-rhizospheric soil samples were collected in three replicates from each cultivation site from July to August in 2019. The rhizosphere soil was stored at −20 °C for analysis of soil bacterial community, and non-rhizosphere soil was sieved and air-dried for analysis of soil chemical properties. The characteristic features of the cultivation site of wild-simulated ginseng were recorded by studying the usual forest physiognomy such as tree species, tree height (TH) and diameter of breast height (DBH), and the topography such as slope direction, slope gradient and height above sea level (HASL) within the stipulated 10 m × 10 m plots of each cultivation site.

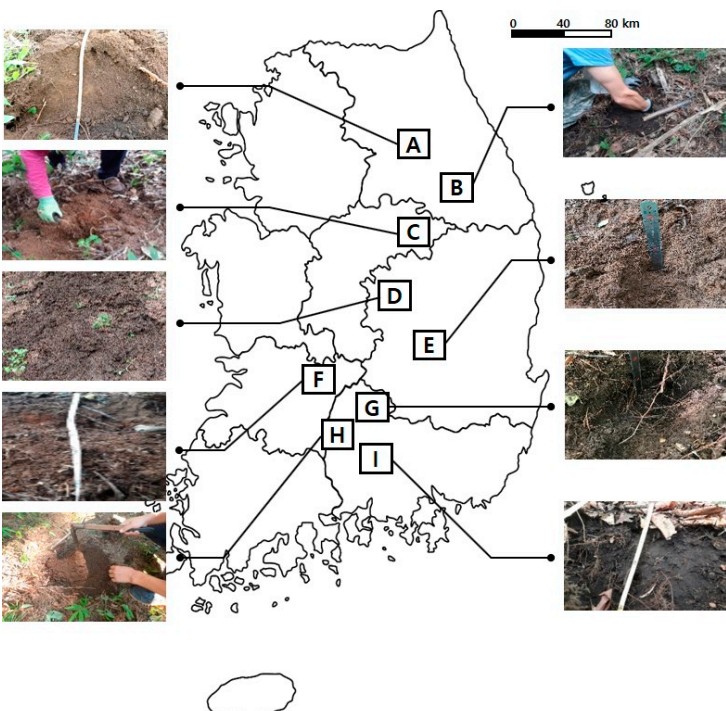

**Figure 1.** Study area and soil sampling sites of 13-year-old Wild-simulated ginseng (WSG). A-I are a wild-simulated ginseng(WSG) cultivation sites in South Korea.

### 2.2. Soil Analysis

Surface soil was removed, and soil was collected at a depth within 20 cm. The soil samples were passed through a 2 mm sieve and air-dried at room temperature. Soil chemical properties analysis was performed following standard analysis manual of the Rural Development Administration (RDA), South Korea [22].

### 2.3. Soil DNA Extraction and PCR Amplification

Total DNA of each rhizosphere soil samples was extracted using DNeasy PowerSoil kit (QIAGEN, Hilden, Germany) following manufacturer instructions. After extraction, quantification and quality of DNA were measured by PicoGreen and Nanodrop (Thermo Scientific, Rockford, IL, USA). Each sequenced sample was prepared according to the Illumina 16S Metagenomics Sequencing Library protocols (Macrogen, Seoul, Korea). In amplicon PCR, V3-V4 region of the 16S rRNA gene of bacteria was targeted using the 16S V3-V4 primers [23]. The 16S V3-V4 primer sequences are as follows: 16S amplicon PCR forward primer, 5′- TCGTCGGCAGCGTCAGATGTGTATAAGAGACAGCCTACGGGNGGCWGC AG-3′, 16S amplicon PCR reverse primer, 5′-GTCTCGTGGGCTCGGAGATGTGTATAAGA GACAGGACTACHVGGGTATCTAATCC -3′. Input gDNA was amplified with 16S V3-V4 primers, and a subsequent limited-cycle amplification step was performed to add multiplexing indices and Illumina sequencing adapters. The conditions for amplicon PCR were as follows. First PCR: initial denaturation at 95 °C for 3 min, followed by 25 cycles of denaturation at 95 °C for 30 s, annealing at 55 °C for 30 s and extension at 72 °C for 30 s, and a final extension at 72 °C for 5 min. The condition for index PCR was as follows. Second PCR: initial denaturation at 95 °C for 3 min, followed by 8 cycles of denaturation at 95 °C for 30 s, annealing at 55 °C for 30 s, and extension at 72 °C for 30 s, and a final extension at 72 °C for 5 min. The final products were normalized and pooled using the PicoGreen, and the size of libraries were verified using the TapeStation DNA screentape D1000 (Agilent, Santa Clara, CA, USA).

### 2.4. Pyrosequencing and Data Processing

Bacterial DNA sequencing was performed using the Illumina MiSeq™ sequencing system (Illumina Inc., San Diego, CA, USA) according to the manufacturer's instructions. Raw sequences of bacterial DNA were processed using Mothur pipeline (version 1.43.0, The University of Michigan, Ann Arbor, MI, USA) [24]. The forward and reverse reads obtained from Illumina platform were assembled, and sequences with the quality score of <20 and the ambiguous nucleotides were discarded before performing downstream analysis. The resulting sequences spanning the V3-V4 region were checked for the presence of chimera using the function chimera.uchime. Taxonomic classification was performed using the "Greengenes reference database" for bacteria. Greengenes was used as it was reported to provide the best combination of speed and quality [25]. The sequences were clustered into operational taxonomic units (OTUs) at 97% similarity level using distance-based greedy clustering method (DGC) in Mothur. OTUs with less than 10 sequences were discarded to reduce false diversity.

### 2.5. Data Analysis

Data are expressed as means $\pm$ standard error (S.E.). Statistical analysis was performed using the program Statistical Analysis System (SAS, version 9.4, SAS Institute, Cary, NC, USA) software for one-way ANOVA and Duncan's test, with statistical significance set at $p < 0.05$ [26]. The data analysis and processing of the 16S rRNA amplicon data was performed following the guidelines [27]. The richness estimators (ACE, Chao and Jackknife) and alpha diversity indices (Shannon and Inverted Simpson) were calculated using Mothur. The principal coordinate analysis (PCoA) was performed using Mothur to visualize the relationship with soil factors based on bacterial community composition. Differences in bacterial community composition were tested using Bray-Curtis dissimilarity values with permutational analysis of variance (PERMANOVA), which is a nonparametric technique

used to differentiate groups based on dissimilarity matrix [28]. DistLM program with 10,000 permutations was used to identify the soil factors explaining the variations in community structure. Correlation coefficient analysis between soil factors and diversity indices were analyzed using Pearson's correlation (IBM SPSS Statistics, version 25, IBM Corp., Armonk, NY, USA).

## 3. Results and Discussion

### 3.1. Location Environment (Topography, Forest Physiognomy, Soil Properties) of the Study Area

The topography and forest physiognomy of WSG cultivation sites are summarized in Table 1. In general, all the cultivation sites were sloped terrain with slope gradient ranging from 5 to 35°; the slope direction varied from east, north, southeast, southwest and northeast; and the sites were 330–920 m above sea level. On the other hand, D and E cultivation sites were identified as broad-leaved forest, and all other cultivation sites were identified as mixed forest of conifer and broad-leaved. Among the cultivation sites, the average TH was maximum in the F cultivation site (26.8 m), and the average DBH was maximum in the A cultivation site (36.1 cm). Furthermore, the chemical properties of soil samples are summarized in Table 2. Soil samples were classified as sandy loam and sandy clay loam based on their soil texture. The soil pH of all cultivation sites has been recorded as acidic soil, and the I cultivation site showed the significantly lowest value compared to other cultivation sites. Organic matter (OM), total nitrogen (TN) contents and cation exchange capacity (CEC) were significantly higher in the A cultivation site, whereas the available phosphate (avail. $P_2O_5$) content was significantly higher in the B cultivation site compared to other cultivation sites. Furthermore, potassium (K) content was recorded at a range of 0.08 to 0.31 $cmol^+ \ kg^{-1}$, calcium (Ca) in the range of 0.10 to 6.99 $cmol^+ \ kg^{-1}$, magnesium (Mg) in the range of 0.05 to 1.07 $cmol^+ \ kg^{-1}$ and sodium (Na) in the range of 0.03 to 0.09 $cmol^+ \ kg^{-1}$, which belongs to the group of exchangeable ions. Forest vegetation is formed by interaction with the environment, and among the forest environments, soil characteristics are majorly affected by the vegetation, and it varies significantly according to the difference in the presence of the particular species of trees [29]. Therefore, the growth and production of WSG cultivated in forest regions have a significant correlation with forest soil and tree species [30]. The organic matter content is higher in broad-leaved forests than in coniferous forests in forest soil because the accumulation of fallen leaves from the trees determines the organic matter content [31,32]. Among the WSG cultivation sites, soil organic matter and total nitrogen content are significantly high in mixed forests with high diversity of deciduous broad-leaved trees [33]. This is because broad-leaved forests contain more organic carbon sources such as fallen leaves than coniferous forests, where organic matter is slowly decomposed [34]. In this study, OM, TN and CEC were high in the WSG cultivation sites with a high percentage broad-leaved tree.

**Table 1.** Location environments of 13-year-old WSG cultivation sites.

| Cultivation Sites | Topography | | | Forest Physiognomy | | | |
| | Slope | | HASL [a] | Species of Tree | Average | | Percentage |
| | | | | | TH [b] | DBH [c] | |
| | ° | Detection | m | | m | cm | % |
| A | 32 | Southeast | 920 | Broad-leaved | 21.5 | 31.0 | 80.0 |
| | | | | Conifer | 36.0 | 56.7 | 20.0 |
| | | | | Total | 24.4 | 36.1 | 100 |
| B | 20 | Southwest | 615 | Broad-leaved | 22.8 | 14.8 | 35.7 |
| | | | | Conifer | 19.0 | 13.0 | 64.3 |
| | | | | Total | 20.4 | 13.7 | 100 |
| C | 30 | Northeast | 387 | Broad-leaved | 14.7 | 12.8 | 81.8 |
| | | | | Conifer | 31.5 | 29.6 | 18.2 |
| | | | | Total | 17.8 | 15.9 | 100 |

**Table 1.** *Cont.*

| Cultivation Sites | Topography | | | Forest Physiognomy | | | |
| | Slope | | HASL [a] | Species of Tree | Average | | Percentage |
| | | | | | TH [b] | DBH [c] | |
| | ° | Detection | m | | m | cm | % |
| D | 20 | North | 530 | Broad-leaved | 18.8 | 15.0 | 100 |
| | | | | Conifer | - | - | - |
| | | | | Total | 18.8 | 15.0 | 100 |
| E | 27 | Southwest | 330 | Broad-leaved | 16.2 | 15.1 | 100 |
| | | | | Conifer | - | - | - |
| | | | | Total | 16.2 | 15.1 | 100 |
| F | 35 | Southeast | 717 | Broad-leaved | 27.3 | 22.8 | 90.9 |
| | | | | Conifer | 22.0 | 23.8 | 9.1 |
| | | | | Total | 26.8 | 22.9 | 100 |
| G | 25 | East | 712 | Broad-leaved | 17.1 | 21.6 | 40.0 |
| | | | | Conifer | 27.0 | 35.6 | 60.0 |
| | | | | Total | 21.7 | 28.0 | 100 |
| H | 15 | North | 743 | Broad-leaved | 14.5 | 18.2 | 40.0 |
| | | | | Conifer | 27.0 | 34.2 | 60.0 |
| | | | | Total | 22.0 | 27.8 | 100 |
| I | 5 | Southeast | 406 | Broad-leaved | 21.0 | 24.8 | 50.0 |
| | | | | Conifer | 20.0 | 36.6 | 50.0 |
| | | | | Total | 20.5 | 30.7 | 100 |

[a] HSAL: height above sea level; [b] TH: tree height; [c] DBH: diameter of breast height.

### 3.2. Bacterial Community Profiles

The bacterial community profiles varied among the soil samples of 13-year-old WSG cultivation sites. The relative abundance of bacterial community at phylum levels is shown in Figure 2. The soil bacterial communities were grouped based on the cultivation sites. *Acidobacteria* (33.6%) was the most dominant phylum in all soil samples, followed by *Proteobacteria* (23.9%), *Verrucomicrobia* (11.2%), *Chloroflexi* (5.9%), *Actinobacteria* (4.4%) and *Planctomycetes* (3.9%). The relative abundance of *Acidobacteria* and *Chloroflexi* was significantly higher in cultivation site I compared to the other cultivation sites, whereas that of *Proteobacteria* was significantly higher in cultivation site F. On the other hand, *Verrucomicrobia*, *Actinobacteria* and *Plantomycetes* were significantly more abundant in cultivation site C. This observation corroborates to previous studies where *Acidobacteria*, *Proteobacteria*, *Verrucomicrobia* and *Actinobacteria* were the major bacterial communities at the phylum level present in soils used for cultivation of *Panax ginseng* [3,35,36]. *Acidobacteria* are acidophilic bacteria mainly present in acidic soils [37]; hence, soil pH is one of the major factors determining *Acidobacteria's* community composition [38–40]. Bacteria belonging to *Acidobacteria* have evolved mechanisms that prefer acidic pH by stabilization of intracellular enzymes [41]. In this study, the relative abundance of *Acidobacteria* was shown to be significantly higher in the cultivation site I, which had the lowest soil pH compared to other studied groups. *Acidobacteria* are shown to be negatively correlated with soil pH in WSG cultivation sites [3], and it was also reported that the *Acidobacteria* population is higher in the cultivated soil of *Panax ginseng* [42,43].



**Table 2.** Soil chemical properties of the samples from 9 different cultivation sites of WSG.

| Cultivation Sites | Soil Texture | pH | EC [a] | OM [b] | TN [c] | Avail. $P_2O_5$ [d] | Exchangeable Cation | | | | CEC [e] |
| | | | | | | | K | Ca | Mg | Na | |
| | | (1:5) | dS m$^{-1}$ | % | % | mg kg$^{-1}$ | cmol$^+$ kg$^{-1}$ | cmol$^+$ kg$^{-1}$ | cmol$^+$ kg$^{-1}$ | cmol$^+$ kg$^{-1}$ | cmol$^+$ kg$^{-1}$ |
|---|---|---|---|---|---|---|---|---|---|---|---|
| A | Sandy clay loam | 4.91 ± 0.07 [ab] | 0.03 ± 0.01 [a] | 17.2 ± 1.18 [a] | 0.69 ± 0.06 [a] | 18.1 ± 4.4 [d] | 0.18 ± 0.08 [ab] | 1.36 ± 0.74 [cd] | 0.35 ± 0.16 [b] | 0.09 ± 0.05 [a] | 36.6 ± 5.2 [a] |
| B | Sandy loam | 5.03 ± 0.07 [a] | 0.02 ± 0.00 [a] | 3.8 ± 0.41 [bc] | 0.15 ± 0.01 [c] | 149.6 ± 9.0 [a] | 0.11 ± 0.01 [b] | 1.76 ± 0.15 [d] | 0.28 ± 0.04 [b] | 0.04 ± 0.00 [a] | 14.3 ± 2.2 [b] |
| C | Sandy clay loam | 5.61 ± 0.14 [a] | 0.02 ± 0.00 [a] | 11.6 ± 0.56 [d] | 0.42 ± 0.01 [d] | 8.6 ± 0.3 [d] | 0.15 ± 0.04 [ab] | 4.70 ± 0.39 [b] | 1.06 ± 0.02 [a] | 0.08 ± 0.02 [a] | 30.9 ± 2.1 [c] |
| D | Sandy loam | 5.61 ± 0.35 [ab] | 0.02 ± 0.00 [a] | 9.5 ± 1.00 [bc] | 0.32 ± 0.05 [bc] | 26.2 ± 8.9 [d] | 0.08 ± 0.01 [b] | 0.54 ± 0.27 [cd] | 0.11 ± 0.06 [b] | 0.06 ± 0.01 [a] | 27.1 ± 2.7 [ab] |
| E | Sandy clay loam | 5.29 ± 0.05 [ab] | 0.05 ± 0.01 [a] | 12.7 ± 0.21 [d] | 0.48 ± 0.01 [d] | 74.1 ± 13.4 [bc] | 0.30 ± 0.04 [a] | 6.99 ± 0.53 [d] | 1.07 ± 0.37 [b] | 0.07 ± 0.02 [a] | 33.5 ± 0.4 [c] |
| F | Sandy loam | 5.11 ± 0.04 [ab] | 0.03 ± 0.00 [a] | 8.5 ± 0.24 [c] | 0.34 ± 0.04 [c] | 105.8 ± 23.2 [b] | 0.19 ± 0.03 [ab] | 2.72 ± 0.77 [c] | 0.38 ± 0.11 [b] | 0.06 ± 0.01 [a] | 27.9 ± 0.9 [b] |
| G | Sandy clay loam | 5.16 ± 0.07 [ab] | 0.02 ± 0.00 [a] | 8.7 ± 0.61 [c] | 0.34 ± 0.01 [c] | 60.5 ± 10.6 [c] | 0.17 ± 0.02 [ab] | 2.03 ± 0.19 [cd] | 0.39 ± 0.11 [b] | 0.05 ± 0.01 [a] | 27.6 ± 0.4 [b] |
| H | Sandy loam | 5.06 ± 0.20 [ab] | 0.05 ± 0.00 [a] | 4.3 ± 0.88 [b] | 0.17 ± 0.04 [b] | 14.8 ± 3.8 [d] | 0.31 ± 0.01 [a] | 0.55 ± 0.16 [a] | 0.16 ± 0.03 [a] | 0.03 ± 0.01 [a] | 14.7 ± 2.2 [ab] |
| I | Sandy loam | 4.73 ± 0.04 [b] | 0.02 ± 0.00 [a] | 8.7 ± 1.06 [c] | 0.34 ± 0.04 [c] | 23.2 ± 9.6 [d] | 0.11 ± 0.02 [b] | 0.10 ± 0.02 [d] | 0.05 ± 0.01 [b] | 0.06 ± 0.02 [a] | 26.0 ± 2.2 [b] |

Each column shows the means of three replications ± standard error (SE). Values in each column with different letters show statistically significant differences ($p < 0.05$) among the treatments according to Duncan's test. [a] EC: electrical conductivity; [b] OM: organic matter; [c] TN: total nitrogen; [d] Avail. P2O5: available phosphorus, [e] CEC: cation exchange capacity.

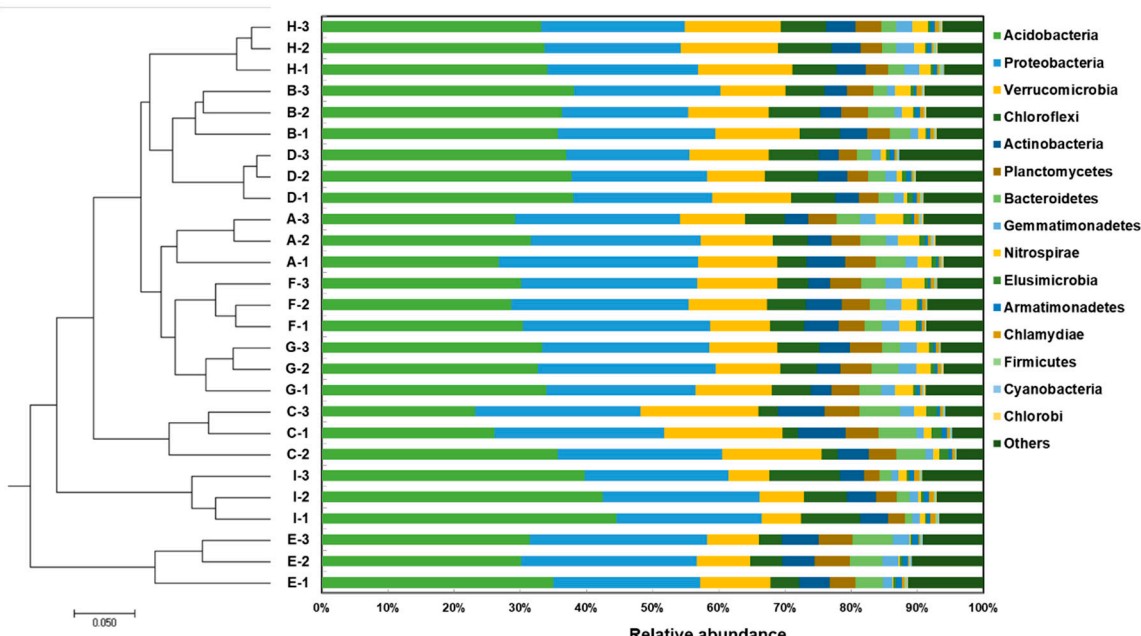

**Figure 2.** The relative abundance of taxonomic composition at the phylum level for bacteria across soil samples, with clustering tree on the left showing the similarities among the soil samples.

### 3.3. Correlation between Soil Bacterial Community and Soil Properties

Principal coordinate analysis (PCoA) and DistLM was done to analyze the correlation between soil bacterial community and the edaphic factors. The two axes of PCoA (Figure 3) explained 45.4% of the total variation in the bacterial community, and soil factors located at each coordinate are related to soil bacterial communities divided by ordinate or abscissa. Soil factors placed in the abscissa are more correlated with soil bacterial communities compared to those located in the ordinate coordinates because the PC1 variation (27.7%) is higher than the variation of PC2 (17.7%). In other words, soil OM, TN, CEC, Mg and Ca affects soil bacterial clustering more than the other soil factors. The DistLM analysis indicated significant correlation between soil factors and soil bacterial community. Cation exchange capacity (CEC), OM, TN, $P_2O_5$, Mg and K were significantly affecting the bacterial community (Table 3). Regarding the sequential tests, the CEC, OM, TN, pH and $P_2O_5$ had a more significant effect on the bacterial community compared to other soil factors. The results of Pearson's correlation analysis between soil factors and diversity indices of bacterial community are represented in Table 4. Among the diversity indices, ace, chao and Shannon diversity index were shown to have a significant positive correlation with OM, TN and CEC. The correlation between soil microbial communities and soil properties had been carried out in numerous studies [44,45]. A study that concentrated on studying the correlation between soil properties and bacterial community in WSG cultivation sites showed that the soil bacterial community is significantly correlated with soil pH, OM, TN and CEC [36]. Soil microorganisms inhabiting the soil have an important relationship with soil quality and productivity such as OM decomposition and nutrient cycling. In addition, the decomposition of OM and nitrogen mineralization in the soil proceeds through a complex interaction of abiotic factors such as soil properties and biotic factors such as microbial population and nutrient demand [46,47]. Soil microorganisms are an important factor affecting soil fertility [48]. The cation exchangeable capacity (CEC) is an indicator of soil fertility and is involved in improving soil buffer capacity, nutrient holding capacity and supplying nutrients [49]. In general, OM, TN and CEC have a high correlation in the natural vegetation [50]. In the results of this study, the soil bacterial community had a

significant correlation with OM, TN and CEC, and this is considered to have a significant correlation with the growth characteristics of WSG.

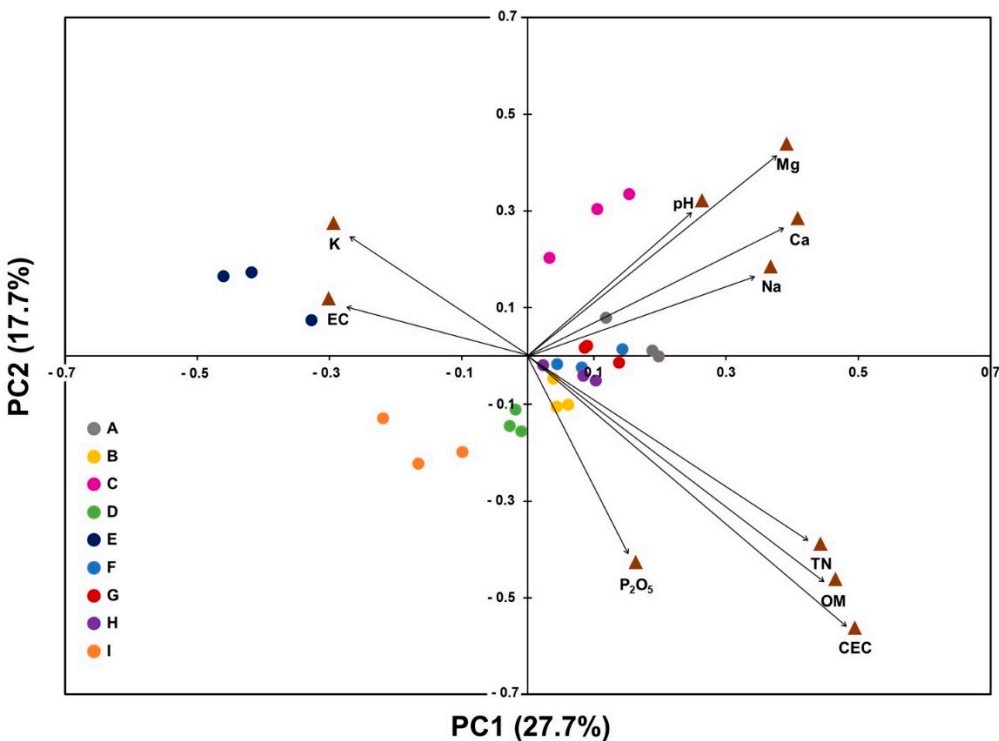

**Figure 3.** Principal coordinate analysis based on Bray–Curtis dissimilarity matrix for the bacterial community generated using themothur platform.

**Table 3.** Marginal and sequential tests of DISTLM on the relation of soil factors variables to the bacterial community of soil samples.

| Soil Factors | Marginal Test | | Sequential Test | | |
|---|---|---|---|---|---|
| | *p*-Value | Proportion | *p*-Value | Proportion | Cumulative |
| CEC | 0.0003 | 0.1410 | 0.0003 | 0.1410 | 0.1410 |
| OM | 0.0011 | 0.1227 | 0.0002 | 0.1089 | 0.2499 |
| TN | 0.0040 | 0.1095 | 0.0017 | 0.1026 | 0.3525 |
| $P_2O_5$ | 0.0425 | 0.0721 | 0.0269 | 0.0522 | 0.4047 |
| pH | 0.0627 | 0.0669 | 0.0297 | 0.0487 | 0.4535 |
| Mg | 0.0078 | 0.1025 | 0.3088 | 0.0290 | 0.4824 |
| K | 0.0522 | 0.0684 | 0.1213 | 0.0367 | 0.5191 |
| Na | 0.2844 | 0.0618 | 0.5460 | 0.0230 | 0.5421 |
| EC | 0.1731 | 0.0509 | 0.6749 | 0.0200 | 0.5620 |
| Ca | 0.2113 | 0.0945 | 0.6489 | 0.0208 | 0.5828 |

**Table 4.** Pearson's correlation analysis between soil factors and diversity indices in cultivation sites of WSG.

| Soil Factors | Correlation Coefficient (*r*) | | | | |
|---|---|---|---|---|---|
| | Ace | Chao | Jackknife | Shannon | Invsimpson |
| pH | 0.225 (0.260) | 0.229 (0.251) | −0.188 (0.348) | 0.228 (0.253) | 0.122 (0.544) |
| EC | −0.074 (0.714) | −0.236 (0.236) | −0.132 (0.512) | −0.234 (0.240) | 0.016 (0.938) |
| OM | 0.459 * (0.016) | 0.380 * (0.050) | −0.115 (0.569) | 0.407 * (0.035) | −0.246 (0.216) |
| TN | 0.425 * (0.027) | 0.371 (0.057) | −0.118 (0.557) | 0.391 * (0.043) | −0.166 (0.408) |
| $P_2O_5$ | −0.343 (0.080) | 0.224 (0.262) | −0.129 (0.521) | 0.248 (0.211) | −0.544 * (0.003) |
| K | 0.038 (0.849) | −0.250 (0.208) | −0.309 (0.116) | −0.264 (0.184) | 0.227 (0.254) |
| Ca | 0.293 (0.138) | 0.260 (0.191) | −0.207 (0.299) | 0.258 (0.193) | 0.255 (0.200) |
| Mg | 0.048 (0.812) | 0.252 (0.204) | −0.201 (0.315) | 0.241 (0.226) | 0.468 * (0.014) |
| Na | 0.111 (0.583) | 0.361 (0.064) | 0.248 (0.213) | 0.366 (0.061) | 0.359 (0.066) |
| CEC | −0.026 (0.899) | 0.434 * (0.024) | −0.052 (0.799) | 0.459 * (0.016) | −0.350 (0.074) |

Correlation coefficients (*r*) are significantly correlated between the variables compared. Negative values denote negative correlation, and positive values denote positive correlation. Values in parentheses refer to *p*-values (* $p < 0.05$).

## 4. Conclusions

The soil bacterial community and diversity of WSG cultivation sites grown in natural conditions in the forest for 13 years had a significant correlation with soil properties such as OM, TN and CEC. Soil pH was recorded to be the most important edaphic factor among the measured soil chemical properties, which drove the abundance of *Acidobacteria* in the studied WSG cultivation sites. This study will enable us to provide a broader idea about the optimum cultivation condition for WSG in natural vegetation condition. In addition, it is believed that more definite information could be provided if a correlation study was conducted on the growth characteristics of WSG and soil bacterial communities according to forest physiognomy and surrounding vegetation along with soil properties.

**Author Contributions:** K.K., K.S.J. and Y.U., conceptualization of the study; K.K. and H.J.K. designed experiments, performed experiments and analyzed sequencing data; H.J.K., D.H.J. and J.H.H. assisted in soil sampling and experiments; H.J.K. assisted in data analysis and discussion; K.K. wrote the manuscript; K.K. and Y.U. performed critical reviewing and editing. All authors have read and agreed to the published version of the manuscript.

**Funding:** This work was supported by the research of National Institute of Forest Science (NIFoS) (FP0802-2017-08).

**Institutional Review Board Statement:** Not applicable.

**Informed Consent Statement:** Not applicable.

**Data Availability Statement:** The data and analyses from the current study are available from the corresponding authors upon reasonable request.

**Conflicts of Interest:** The authors declare no conflict of interest.

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
