# Peer review of "Correlation between Soil Bacterial Community Structure and Soil Properties in Cultivation Sites of 13-Year-Old Wild-Simulated Ginseng (Panax ginseng C.A. Meyer)"

_applsci, doi:10.3390/app11030937_

Round 1
Reviewer 1 Report
In general, the authors have done a good job explaining the background information necessary to appreciate the rationale and results of the experiments. The manuscript was prepared correctly. Methodology and analysis of results rather don't raise any objections.
However, some minor amendments are needed. The discussion of the results is written a bit generally. There are papers related to this topic that the authors did not cite. An assessment putting the findings into perspective and make a solid conclusion is missing. The authors should emphasize more the novelty and usefulness of the results.
Author Response
Thank you for reviewer’s deep review comments. We would like to extend gratitude towards the reviewers for their comments on the manuscript. The detailed response to the reviewer’s comments is enlisted below:
Reviewer 1
Comment
In general, the authors have done a good job explaining the background information necessary to appreciate the rationale and results of the experiments. The manuscript was prepared correctly. Methodology and analysis of results rather don't raise any objections. However, some minor amendments are needed. The discussion of the results is written a bit generally. There are papers related to this topic that the authors did not cite. An assessment putting the findings into perspective and make a solid conclusion is missing. The authors should emphasize more the novelty and usefulness of the results
Response
A bit more specific references were added in the discussion part to make more specific description about the results obtained in the given study. Based on the modifications, abstract and conclusion sections were edited for giving better overview.
Reviewer 2 Report
The manuscript entitled "Correlation Between Soil Bacterial Community Structure and Soil Properties in Cultivation Sites of 13-year-old Wild-simulated Ginseng (Panax ginseng C.A. Meyer)" investigated the soil bacterial community structure in cultivation sites of 13-year-old wild-simulated ginseng and how the soil properties influence the bacterial communities. The authors examined the soil chemical properties and bacterial communities in ginseng soils. They argued that their results showed that Acidobacteria were the most dominant phyla in WSG cultivation sites and the total organic matter, total nitrogen and cation exchange capacity were important factors of influencing the bacterial community. This article is interesting and should be of interest to the readers of Applied Sciences. I recommend acceptance after addressing my following comments for publication.
Comments
Abstract:
- The Abstract looks like a part of Introduction and some sentences seem redundant, but Abstract need to be concise and give vital information. Please rephrase the background information and concisely show the reasons and value of this research.
- The abstract lacks convincing results and conclusion. Please consider adding more interesting results of the experiments and add one or two more sentences to address the connection between the results of soil properties structuring soil bacterial community and the cultivation and productivity of WSG.
Introduction:
- Lines 31-32: " Soil microbes present in the rhizosphere has symbiotic relation with plants, and it helps in plant growth"
Comment: "Soil microbes present in the rhizosphere have symbiotic relations with plants, and they can contribute to plant growth"
- Lines 34-36: "The recent advent in development of culture-independent methods has made it convenient to study microbial diversity and predict key functional traits of soil microbiota [6]."
The use of “advent” and “development” seems conflicting. Please rephrase.
- Lines 36-37: “There is myriad of environmental factors which can affect the soil properties…”
Comment: “Myriad of environmental factors can affect the soil properties…”
- Lines 38-39: "Therefore, studying the correlation of environmental factors and soil microbial community is very important [11]."
Comment: Add more references for this conclusion sentence. Please consider:
Liang, X., Zhang, Y., Wommack, K.E., Wilhelm, S.W., DeBruyn, J.M., Sherfy, A.C., Zhuang, J. and Radosevich, M., 2020. Lysogenic reproductive strategies of viral communities vary with soil depth and are correlated with bacterial diversity. Soil Biology and Biochemistry, p.107767.
Roy, K., Ghosh, D., DeBruyn, J.M., Dasgupta, T., Wommack, K.E., Liang, X., Wagner, R.E. and Radosevich, M., 2020. Temporal dynamics of soil virus and bacterial populations in agricultural and early plant successional soils. Frontiers in Microbiology, 11, 1494.
- Line 46-47: “However, the correlation between soil microbial communities and edaphic factors exerted on medicinal crops growing in forest are insufficient.”
Comment: “However, the correlation between soil microbial communities and edaphic factors exerted on medicinal crops growing in forest is insufficiently studied.”
Materials and Methods
- Line 59: “and the details of the study area and the sampling site are shown in Figure 1”
Comment: “and the details of the study area and the sampling sites are shown in Figure 1”
- Line 61: "during July to August 2019"
Suggestion: “from July to August in 2019". Please also specify the exact time for each sampling effort.
- Lines 82-83: "Each sequenced sample is prepared according to the Illumina 16S Metagenomics Sequencing Library protocols "
Comment: "Each sequenced sample was prepared according to the Illumina 16S Metagenomics Sequencing Library protocols "
- Line 90: Change “is” to “was”
The phage titre in the presence of QS inhibitor was higher than in the control. Therefore, does the decrease in the amount of phage removed by CRISPR mean that the phages have higher efficacy for infection? This is opposite to your conclusion, but I think my understanding is wrong. Please explain the relationship among CRISPR removement of phage, phage titre, and phage efficacy.
- Line 96-98: "The final products are normalized and pooled using the PicoGreen, and the size of libraries are verified using the TapeStation DNA screentape D1000 (Agilent, Santa Clara, CA, USA)."
Comment: Change “are” to “were”
Results and Discussion
- Line 170: “3.2. Bacterial community profiels”
Comment: “3.2. Bacterial community profiles”
Suggestion: “The increased fitness of CRISPR immune bacteria further supports the idea that they are infected at a lower frequency when QS is inhibited.”
- Line 172-173: “Relative abundance of bacterial community at phylum levels are shown in Figure 2.”
Comment: Change “relative abundance” to “The relative abundances”
- “The soil bacterial community grouped based on the cultivation sites.”
Comment: “The soil bacterial communities grouped based on the cultivation sites.”
- Line 177-178: “and it was also reported that it hashigh relative abundance in the cultivated soil of Panax ginseng [32, 33].”
Comment: “and it was also reported that there were high relative abundances of Acidobacteria in the cultivated soil of Panax ginseng [32, 33].”
Please also discuss the relation of high relative abundances of Acidobacteria with the soil pH in your study. The relative abundances of other bacterial phyla, e.g., Proteobacteria and Verrucomicrobia were also high in your samples. I think it might also be worth some discussion. You did not discuss the differences in the relative abundance of the bacterial phyla between different sampling sites. The readers might want to see what bacterial groups were driving the differences of bacterial community profiles.
- In describing your results, I noticed you used both present tense and past tense. I think most researchers use past tense to show their results, as the experiments were performed in the past. Please be consistent.
Author Response
Thank you for reviewer’s deep review comments. We would like to extend gratitude towards the reviewers for their comments on the manuscript. The detailed response to the reviewer’s comments is enlisted below:
Reviewer 2
Comment 1
Abstract looks like a part of Introduction and some sentences seem redundant, but Abstract need to be concise and give vital information. Please rephrase the background information and concisely show the reasons and value of this research
Response
The abstract was modified and more specific information was given based on the study.
Comment 2
Abstract lacks convincing results and conclusion. Please consider adding more interesting results of the experiments and add one or two more sentences to address the connection between the results of soil properties structuring soil bacterial community and the cultivation and productivity of WSG.
Response
The important information based on the results have been added on the abstract
Comment 3
Lines 31-32: " Soil microbes present in the rhizosphere has symbiotic relation with plants, and it helps in plant growth" Comment: "Soil microbes present in the rhizosphere have symbiotic relations with plants, and they can contribute to plant growth"
Response
The error in line 27-28 was corrected in the manuscript.
"Soil microbes present in the rhizosphere have symbiotic relations with plants, and they can contribute to plant growth
Comment 4
Lines 34-36: "The recent advent in development of culture-independent methods has made it convenient to study microbial diversity and predict key functional traits of soil microbiota [6]. "The use of “advent” and “development” seems conflicting. Please rephrase.
Response
The error in line 34-36 was corrected in the manuscript.
The recent develop in culture-independent methods has made it convenient to study mi-crobial diversity and predict key functional traits of soil microbiota [6].
Comment 5
Lines 36-37: “There is myriad of environmental factors which can affect the soil properties…” Comment: “Myriad of environmental factors can affect the soil properties…”
Response
The error in line 36-37 was corrected in the manuscript
Myriad of environmental factors can affect the soil properties and in turn tweak diversity and composition of soil microbiota [8, 9].
Comment 6
Lines 38-39: "Therefore, studying the correlation of environmental factors and soil microbial community is very important [11]." Comment: Add more references for this conclusion sentence. Please consider:
Response
Revised to reflect the reviewer comment and added references.
Noll, M.; Wellinger, M. Changes of the soil ecosystem along a receding glacier: Testing the correlation between envi-ronmental factors and bacterial community structure. Soil Biology and Biochemistry 2008, 40, 2611-2619.
Gao, P.; Xu, W.; Songtag, P.; Li. X.; Xue, G.; Liu, T.; Sun, W. Correlating microbial community compositions with envi-ronmental factors in activated sludge from four full-scale municipal wastewater treatment plants in Shanghai, China. Environmental Biotechnology 2016, 100, 4633-4673.
Liang, X., Zhang, Y., Wommack, K.E., Wilhelm, S.W., DeBruyn, J.M., Sherfy, A.C., Zhuang, J. and Radosevich, M., 2020. Lysogenic reproductive strategies of viral communities vary with soil depth and are correlated with bacterial diversity. Soil Biology and Biochemistry, p.107767.
Roy, K., Ghosh, D., DeBruyn, J.M., Dasgupta, T., Wommack, K.E., Liang, X., Wagner, R.E. and Radosevich, M., 2020. Temporal dynamics of soil virus and bacterial populations in agricultural and early plant successional soils. Frontiers in Microbiology, 11, 1494.
Comment 7
Line 46-47: “However, the correlation between soil microbial communities and edaphic factors exerted on medicinal crops growing in forest are insufficient.” Comment: “However, the correlation between soil microbial communities and edaphic factors exerted on medicinal crops growing in forest is insufficiently studied.”
Response
The error in line 46-47 was corrected in the manuscript.
“However, the correlation between soil microbial communities and edaphic factors exerted on medicinal crops growing in forest is insufficiently studied.”
Comment 8
Line 59: “and the details of the study area and the sampling site are shown in Figure 1” Comment: “and the details of the study area and the sampling sites are shown in Figure 1”
Response
The error in line 59 was corrected in the manuscript.
and the details of the study area and the sampling sites are shown in Figure 1.
Comment 9
Line 61: "during July to August 2019" Suggestion: “from July to August in 2019". Please also specify the exact time for each sampling effort.
Response
The error in line 59 was corrected in the manuscript.
Both rhizospheric and non-rhizospheric soil samples were collected in three replicates from each cultivation sites from July to August in 2019.
Comment 10
Lines 82-83: "Each sequenced sample is prepared according to the Illumina 16S Metagenomics Sequencing Library protocols " Comment: "Each sequenced sample was prepared according to the Illumina 16S Metagenomics Sequencing Library protocols "
Response
The error in line 82-83 was corrected in the manuscript.
Each sequenced sample was prepared according to the Illumina 16S Metagenomics Se-quencing Library protocols
Comment 11
Line 90: Change “is” to “was”
Response
The error in line 90 was corrected in the manuscript.
Input gDNA was amplified with 16S V3-V4 primers and a subsequent limited‐cycle am-plification step was performed to add multiplexing indices and Illumina sequencing adapters.
Comment 12
Line 96-98: "The final products are normalized and pooled using the PicoGreen, and the size of libraries are verified using the TapeStation DNA screentape D1000 (Agilent, Santa Clara, CA, USA)." Comment: Change “are” to “were”
Response
The error in line 96-98 was corrected in the manuscript.
The final products are normalized and pooled using the PicoGreen, and the size of librar-ies were verified using the TapeStation DNA screentape D1000 (Agilent, Santa Clara, CA, USA).
Comment 13
“3.2. Bacterial community profiels” Comment: “3.2. Bacterial community profiles
Response
The error in subtitle was corrected in the manuscript.
“3.2. Bacterial community profiles”
Comment 14
Line 172-173: “Relative abundance of bacterial community at phylum levels are shown in Figure 2.” Comment: Change “relative abundance” to “The relative abundances”
Response
The error in line 172-173 was corrected in the manuscript
The relative abundance of bacterial community at phylum levels was shown in Figure 2.
Comment 15
“The soil bacterial community grouped based on the cultivation sites.” Comment: “The soil bacterial communities grouped based on the cultivation sites.”
Response
The error in line 173 was corrected in the manuscript
The soil bacterial communities grouped based on the cultivation sites.
Comment 16
Line 177-178: “and it was also reported that it has high relative abundance in the cultivated soil of Panax ginseng [32, 33].” Comment: “and it was also reported that there were high relative abundances of Acidobacteria in the cultivated soil of Panax ginseng [32, 33].
Response
The error in line 177-178 was corrected in the manuscript
and it was also reported that the Acidobacteria population is higher in the cultivated soil of Panax ginseng [42, 43].
Comment 17
Please also discuss the relation of high relative abundances of Acidobacteria with the soil pH in your study.
Response
According to the comment of reviewer, added more discussion for relation between relative abundance of Acidobacteria and soil pH
Comment 18
The relative abundances of other bacterial phyla, e.g., Proteobacteria and Verrucomicrobia were also high in your samples. I think it might also be worth some discussion. You did not discuss the differences in the relative abundance of the bacterial phyla between different sampling sites. The readers might want to see what bacterial groups were driving the differences of bacterial community profiles.
Response
According to the comment of reviewer, statistical results and discussion were added for the relative abundance of bacteria between sampling sites.
Comment 19
In describing your results, I noticed you used both present tense and past tense. I think most researchers use past tense to show their results, as the experiments were performed in the past. Please be consistent
Response
Revised to reflect the reviewer comment.
Reviewer 3 Report
The article study concerns on deciphering the correlation between soil chemical properties and bacterial community profiles of WSG plants. The results schould helps to identify the suitable cultivation sites for WSG and increase the productivity of these plants.
The design of the study involving providing data in real condition makes the results valuable. Therefore I recommend publication with some comments below that may need in some parts major revisions.
I observe that from the beginning there is no information about available of statistical methods, that could helps to correlate the resulds of research. Ahe authors do not detail the available statistical possibilities. An example is the ANOVA method with Duncan's test. It is worth mentioning, and refers the literature source, that there are other methods than just Pearson’s correlation, for example see
Nurek, T., Gendek, A., & Roman, K. (2018). Forest Residues as a Renewable Source of Energy: Elemental Composition and Physical Properties. BioResources, 14(1), 6-20. Retrieved from https://ojs.cnr.ncsu.edu/index.php/BioRes/article/view/BioRes_14_1_6_Nurek_Forest_Residues_Renewable_Energy/6480
Author Response
Thank you for reviewer’s deep review comments. We would like to extend gratitude towards the reviewers for their comments on the manuscript. The detailed response to the reviewer’s comments is enlisted below:
Reviewer 3
Comment
I observe that from the beginning there is no information about available of statistical methods, that could help to correlate the results of research. The authors do not detail the available statistical possibilities. An example is the ANOVA method with Duncan's test. It is worth mentioning, and refers the literature source, that there are other methods than just Pearson’s correlation, for example see
Response
In reflection of the reviewer comments, a SAS program (Duncan’s test) was used to add the results of significant difference between the cultivation sites for soil properties and relative abundance of soil bacterial community.
Round 2
Reviewer 3 Report
The authors adapted the article to the comments of the reviewer and divide data to the homogeneous groups in Table 2. Post-hoc test (Duncan's test) was included, but to make it fully understandable for other readers, I recommend using the reference in line 120-123 for literature, e.g.
Nurek, T.; Gendek, A.; Roman, K. Forest residues as a renewable source of energy: Elemental composition and physical properties. BioResources 2019, 14, 6–20. https://ojs.cnr.ncsu.edu/index.php/BioRes/article/view/BioRes_14_1_6_Nurek_Forest_Residues_Renewable_Energy/6480
The above article does not explain the operation of ANOVA, but it can always bring a little closer to the fact that similar analyzes have already been carried out in relation for the natural environment, biomass, soil, etc.
Author Response
Thank you for reviewer’s deep review comments. We would like to extend gratitude towards the reviewers for their comments on the manuscript. The detailed response to the reviewer’s comments is enlisted below:
Reviewer 3
Comment
The authors adapted the article to the comments of the reviewer and divide data to the homogeneous groups in Table 2. Post-hoc test (Duncan's test) was included, but to make it fully understandable for other readers, I recommend using the reference in line 120-123 for literature, e.g.
Nurek, T.; Gendek, A.; Roman, K. Forest residues as a renewable source of energy: Elemental composition and physical properties. BioResources 2019, 14, 6–20. https://ojs.cnr.ncsu.edu/index.php/BioRes/article/view/BioRes_14_1_6_Nurek_Forest_Residues_Renewable_Energy/6480
The above article does not explain the operation of ANOVA, but it can always bring a little closer to the fact that similar analyzes have already been carried out in relation for the natural environment, biomass, soil, etc.
Response
Revised to reflect the reviewer comment and added references.
Nurek, T.; Gendek, A.; Roman, K. Forest residues as a renewable source of energy: Elemental composition and physical properties. BioResources 2019, 14, 6–20.
